# Cardiovascular Function in Different Phases of the Menstrual Cycle in Healthy Women of Reproductive Age

**DOI:** 10.3390/jcm11195861

**Published:** 2022-10-03

**Authors:** Malgorzata Kwissa, Tomasz Krauze, Agnieszka Mitkowska-Redman, Beata Banaszewska, Robert Z. Spaczynski, Andrzej Wykretowicz, Przemyslaw Guzik

**Affiliations:** 1Department of Cardiology-Intensive Care Therapy and Internal Medicine, Poznan University of Medical Sciences, 61-701 Poznan, Poland; 2Department of Internal Medicine, Poznan University of Medical Sciences, 61-701 Poznan, Poland; 3Division of Infertility and Reproductive Endocrinology, Department of Gynaecology, Obstetrics, and Gynaecological Oncology, Poznan University of Medical Sciences, 61-701 Poznan, Poland; 4Center for Gynecology, Obstetrics and Infertility Pastelowa, 60-198 Poznan, Poland

**Keywords:** blood pressure, body water content, hemodynamic function, menstrual cycle, sex hormones

## Abstract

Background: Sex hormones influence the cardiovascular (CV) function in women. However, it is uncertain whether their physiological variation related to the regular menstrual cycle affects the CV system. We studied changes in the hemodynamic profile and body’s water content and their relation to sex hormone concentration in healthy women during the menstrual cycle. Material and methods: Forty-five adult women were examined during the early follicular, late follicular, and mid-luteal phases of the same menstrual cycle. The hemodynamic profile was estimated non-invasively by cardiac impedance while water content was estimated by total body impedance. Results were compared with repeated measures ANOVA with post-test, if applicable. Results: There were no significant changes in most hemodynamic and water content parameters between the menstrual cycle phases in healthy women. Left ventricular ejection time differed significantly among phases of the menstrual cycle, with shorter values in the mid-luteal phase (308.4 vs. 313.52 ms, *p* < 0.05) compared to the late follicular phase. However, the clinical relevance of such small differences is negligible. Conclusions: Changes in sex hormones during the physiological menstrual cycle appear to have no considerable effect on healthy women’s hemodynamic function and water accumulation.

## 1. Introduction

Some of the actions of sex hormones (progesterone, estrogens, and androgens) involve the cardiovascular system. Estrogen and progesterone receptors are present in the cardiovascular system [1,2]. Estrogen modulates the tone of the blood vessel wall. It enhances the production of vasodilatory mediators and antioxidant enzymes by endothelium. By increasing the vascular expression of beta-adrenergic receptors, estrogen causes the vascular muscle cells relaxation [2,3,4]. In animal studies, long-term estrogen deficiency correlated with decreased contractility of cardiomyocytes [5]. Estrogen regulates ion currents in cardiomyocytes by reducing the excitation–contraction coupling gain and prolonging their action potential time [1,5,6,7]. Progesterone dilates vessels through an increase in the prostacyclin synthesis [8], inhibits calcium, and enhances potassium current in the myocardial cells causing shortening of the QT interval [9].

Sex steroids also regulate body fluids management. The kidney tubules have estrogen receptors [2]. Both estrogen and progesterone impact renin-angiotensin system activity [10,11] and renal sodium reabsorption [12]. Estrogen lowers the osmotic threshold for vasopressin release in the hypothalamus. Progesterone competes with aldosterone for the same mineralocorticoid receptor causing transient natriuresis [13]. During pregnancy, concentrations of sex hormones increase, which leads to expanding circulating blood volume and significant hemodynamic changes. Additionally, systemic vascular resistance and blood pressure (BP) primarily decline, followed by a compensatory heart rate (HR) acceleration. Typically, the stroke volume and cardiac output increase in pregnant women [14,15].

The hormonal changes present during the menstrual cycle are dynamic yet less pronounced than during pregnancy. In contrast to studies on pregnant women, data on hemodynamic alterations during the physiological menstrual cycle are sparse and inconclusive [16,17,18,19].

Our study compared hemodynamic variables during the same physiological menstrual cycle of healthy women. For this purpose, we used non-invasively measured hemodynamic parameters describing various functions of the heart and circulation and indices of body water content. We expect significant changes in the values of the variables used. Such observation would have important clinical implications, as it should influence the interpretation of the clinical condition of women of reproductive age depending on the menstrual cycle phase.

## 2. Materials and Methods

It is a post-hoc analysis of a prospective study conducted between 2009 and 2013 [19]. The Bioethics Committee at Poznan University of Medical Sciences in Poland approved the original prospective project’s study protocol [no 665/09]. At the time of enrollment, all women gave written informed consent for their participation. No additional approval from the Bioethical Committee was required for this retrospective analysis. All data analyses for this study were performed anonymously.

### 2.1. Participants

Calls for recruitment were made through announcements and educational actions at local universities in Poznan, Poland. In the original study [19], the following inclusion criteria were defined: volunteer participation; reproductive age; regular menstruations (27–31 days); no hormonal therapy or hormonal contraception for at least two previous months; and no clinical signs and symptoms of endocrinopathy with a particular focus on hyperandrogenemia, hyperandrogenism, hyperprolactinemia, thyroid disease, and normal resting ECG. The following excluding factors were executed: current smoking, history of hypertension, diabetes mellitus, cardiovascular, renal, liver, and neoplastic disease. During this project, forty-five women fulfilled the enrolment criteria. None of the women were active smokers.

### 2.2. Study Protocol

All participants refrained from eating or drinking coffee or caffeine-containing products. Each woman underwent body mass and non-invasive cardiovascular measurement three times during the same menstrual cycle, i.e.,

-Early follicular phase (EFP)—between the third and fifth day;-Late follicular phase (LFP)—between the 11th and 14th day;-Mid-luteal phase (MLP)—between the 19th and 22nd day.

To confirm the physiological course of the menstrual cycle, a gynaecological examination, including a transvaginal ultrasound scan (Aloka ProSound alpha7, Aloka Co., Ltd., Tokyo, Japan), was carried out at all three phases. The ultrasound evaluated the size and number of growing ovarian follicles, confirmed the presence of the corpus luteum, and assessed changes in the endometrial echogenicity and thickness. Concentrations of estradiol, LH, and FSH (automated Cobas e601 immunoanalyser, Roche, Warsaw, Poland) were measured in venous plasma samples in each phase and progesterone in the mid-luteal phase [20].

### 2.3. Measurements

#### 2.3.1. Non-Invasive Assessment of the Cardiovascular Signals

Brachial blood pressure was measured automatically by the M5 Blood Pressure Monitor (Omron Healthcare, Kyoto, Japan) on both arms in sitting participants. Next, all women had put on:-Four pairs of single-use Ag/AgCl electrodes, placed on both sides of the chest (at the crossings of the midaxillary lines with the level with the end of the xiphoid process—lower pairs) and the neck (at the crossings of the base of the neck with the prolongation of the midaxillary lines—upper pairs). These electrodes were connected to the cardiac impedance monitor Niccomo (Medis GmbH, Ilmenau, Germany);-A pair of single-use Ag/AgCl electrodes for continuous ECG recorded by the analogue/digital converter Porti 7 (TMSI, Oldenzaal, The Netherlands) for online monitoring of the presence of arrhythmia on a PC monitor.

Participants were resting in a supine position for 15 min for cardiovascular adaptation. After this time, we recorded all the signals for the next 5 min.

#### 2.3.2. Haemodynamic Parameters by Cardiac Impedance

The cardiac impedance signal was continuously recorded for 5 min by the Niccomo monitor for the following hemodynamic parameters (according to the modified Bernstein formula) [21]:-SI—stroke index, i.e., the stroke volume normalised to the body surface area;-CI—cardiac index, i.e., the cardiac output normalised to the body surface area;-SVRI—systemic vascular resistance index, i.e., the systemic vascular resistance normalised to the body surface area;-HR—heart rate;-LVET—left ventricular ejection time;-PEP—pre-ejection period;-STR—PEP to LVET ratio.

SVRI reflects the changes in afterload, while the SI and CI depend on the amount of venous return and preload, myocardial contractility, and afterload. CI nearly always equals venous return in healthy people and reflects the amount of circulating blood [22].

The PEP is the electromechanical delay associated with the electrical depolarisation of the ventricles and the isovolumetric contraction time of the left ventricle (LV). Higher sympathetic activity increases LV contractility and shortens PEP [23,24]. LVET is the time when LV ejects blood to the aorta—it is measured between the opening and closing of the aortic valve. The LVET shortening suggests a reduced SV or increased afterload [23]. The STR increases with impaired LV systolic function [25].

#### 2.3.3. Indirect Measures of Water Accumulation

Water is the most significant contributor to body mass—we measured body mass during all menstrual cycle phases to reflect its content. We also estimated the thoracic fluid content (TFC) by cardiac impedance. TFC is proportional to the amount of fluid in the chest as an index of regional water accumulation in the lungs. Additionally, we used an eight-electrode total body impedance analyser (Tanita MC 180 MA, Tanita, Tokyo, Japan) to measure the amount of total body water and its relative contribution to body weight. This device uses low electrical currents at four frequencies, i.e., 5, 50, 250, and 500 kHz [26].

### 2.4. Statistical Analysis

Data distribution was not normal (D’Agostino–Pearson test). Thus, continuous data are presented as median and the 25th–75th percentiles. For the trend analysis (either linear or cubic) between the measured values of all parameters and different phases of the menstrual cycle, we applied repeated measures Friedman test (nonparametric ANOVA) with post-tests. The MedCalc^®^ Statistical Software version 20.110 (MedCalc Software Ltd., Ostend, Belgium; Available online: https://www.medcalc.org, accessed on 11 June 2022) was employed for statistical analysis. Only *p* values < 0.05 were considered significant.

## 3. Results

The median age of the studied women was 28 (interquartile range 26–34), and their BMI, WHR, fasting glucose, insulin, and lipid fractions concentrations were within normal ranges. Table 1 summarises the hormonal profile and clinical characteristics of the studied women during the menstrual cycle.

Table 2 shows changes in the haemodynamics and indices of water amount across different phases of the same physiological menstrual cycle. The repeated measures ANOVA revealed no statistically significant difference in the values of the analysed indirect indices of water content and hemodynamic parameters, except for LVET.

Pairwise post-hoc analysis showed that the LVET value is significantly greater in LFP than MLP (Figure 1).

## 4. Discussion

Neither most haemodynamic parameters nor indirect body water content indices change in healthy women in different phases of the menstrual cycle. Although LVET significantly changes over the menstrual cycle, mainly between the LFP and MLP, the magnitude of the difference is subtle and somewhat of no clinical meaning. The difference in LVET medians between these two phases is 5.12 ms, which is 1.63% of the median LVET during LFP.

Sex hormones influence many cells, tissues, organs, and systems. The cardiovascular system is an example. Effects of sex hormones vary depending on their concentrations, which fluctuate during normal menstrual cycles and pregnancy. A gradual rise in sex steroids in pregnancy correlates with increases in CI and SI and decreases in SVRI and blood pressure [14,15,27,28].

In healthy non-pregnant women, changes in estrogen and progesterone concentrations during the whole menstrual cycle are relatively small compared to pregnancy [14,28]. Chapman et al. found that cardiac output was higher in the mid-luteal compared with the mid-follicular phase and explained their findings by an increased plasma renin activity and aldosterone concentration [29]. Aldosterone expands the intravascular fluid and might be the first hemodynamic adaptation to a potential pregnancy. However, Chapman et al. studied only 16 women who had indirectly confirmed ovulation by identifying the LH surge in urine—the method is less sensitive and precise for ovulation identification than ultrasonography. In contrast, we visualised the growing follicle and corpus luteum by ultrasound and measured the progesterone concentration in the blood [20]. So far, no other study has demonstrated similar hemodynamic changes during the menstrual cycle as Chapman et al. In another study, Gordon et al. analysed the cardiovascular response by cardiac impedance to mental stress during the menstrual cycle in 57 young, healthy women [30]. They found no significant differences in SI and CI in different phases of the menstrual cycle at rest.

In addition to aldosterone, physiological concentrations of sex hormones increase fluid body content [31]. However, it is uncertain whether the intravascular fluid content changes during the menstrual cycle. Information on intravascular water can be derived from studying CI, i.e., the amount of circulating blood. CI is a product of heart rate and SI (the amount of blood ejected from the left ventricle during a single contraction). According to the Frank–Starling heart law, SI increases in response to the extension of the left ventricular size (and its end-diastolic volume) [22].

In our study, neither CI nor SI changed during the menstrual cycle, suggesting that the intravascular water content remains stable over its various phases. Additionally, all other indices of water content, i.e., total body mass, total and relative (percentage) body water content, and the regional water content in the lungs (TFC), did not change over the menstrual cycle. Cumberledge et al. presented similar findings on the percentage of total body water. Additionally, they reported that extracellular and intracellular compartments of the total body water did not alter in healthy young women throughout the menstrual cycle [32].

Earlier studies postulated water shifts between body compartments with a significant change in the intravascular volume during the menstrual cycle. Stachenfeld et al. reported that estrogen from transdermal patches increased plasma volume but reduced the extracellular fluid volume. They speculated that it might be attributable to protein diffusion across the capillary endothelium, an adaptive change in the plasma, and oncotic tissue pressures. Administration of combined estrogen and progesterone increased plasma volume but did not influence the extracellular fluid, probably due to the additional sodium and water maintenance [31]. Progesterone increases the capillary permeability by allowing fluid and albumin to cross into the interstitial space [33]. Hulde et al. found that the blood concentration of the endothelial glycocalyx forming a barrier to the passage of proteins through the vascular membranes was higher in the luteal phase and lower in the periovulatory period. These results indicate that progesterone contributes to the degradation of the endothelial glycocalyx and impacts vascular permeability [34].

Some studies show significant changes in the resting heart rate during the menstrual cycle. Moran et al. noticed an increase in heart rate in the ovulatory and luteal phases compared to the follicular phase [35]. Brar et al. demonstrated the highest HR in the luteal and the lowest HR in the periovulatory phase. [36]. However, we did not observe the difference in heart rate during the menstrual cycle, and our observation corresponds with other reports [37,38].

Potential differences in HR related to the menstrual cycle have been explained by changes in the sympatho-vagal balance, sympathetic activity, or parasympathetic tone [36]. Such conclusions, however, come from studies using heart rate variability (HRV) parameters. Unfortunately, there are many critical issues with HRV parameters as indices of the autonomic control of the heart rate. Over 20 years ago, HRV was already suggested as not a good proxy of the sympathovagal balance or vagal tone [39,40]. Recently, Marmerstein et al., based on interesting animal studies in rats, have provided evidence that HRV parameters assumed to reflect the vagal tone are not correlated with vagal activity at all [41]. Moreover, Brar et al., who studied HR and HRV during the menstrual cycle, used the so-called normalized power in the low (LFnu) and high (HFnu) frequencies and the LF to HF ratio (LF/HF) as indices of vagal activity. They did not notice, however, that both LFnu and HFnu add up to 100% and are mirroring one another. Moreover, one mathematical equation can easily derive the LF/HF from either LFnu or HFnu. In other words, physiological (if at all possible) interpretation of LFnu, HFnu, and LF/HF is identical. Therefore, no conclusions on cardiac autonomic control across the menstrual cycle in healthy women should be drawn from HRV.

Undoubtedly, many studies have shown that HRV may change during various phases of the menstrual cycle. Nevertheless, it is uncertain whether HRV indices can be explained by the modulation of cardiac autonomic control alone. HRV, as a derivative of HR, depends de facto on many more physiological and non-physiological factors and internal and external influences.

Systolic time intervals, i.e., PEP, LVET, and their STR ratio, indirectly describe LV systolic function. Increased STR, shortened LVET, and prolonged PEP were observed in cardiac patients with systolic dysfunction [42]. SI is directly derived from LVET and strongly depends on LV contractility. As SI, PEP, and STR did not differ among various menstrual cycle phases, it suggests that LV contractility is not affected. Although LVET was statistically shorter in the MLP than LFP, this difference has no practical meaning. Additionally, after LVET was normalized to the cardiac cycle duration, this difference disappeared (*p* = 0.67 by the Friedman test).

LV systolic function was studied by echocardiography during the menstrual cycle and pregnancy. Fuenmayor et al. showed no significant differences in the ejection fraction between the follicular and luteal phases in healthy women [43]. Grzybowski et al. found no significant menstrual-cycle-phase-dependent differences in LV myocardial contractility at rest or during exercise echocardiography in early and late follicular phases in women with angina pectoris without coronary artery stenosis [44]. Most studies on pregnant women showed no changes in the ejection fraction [45,46]. However, some reported a reduction in LV contractile function at the end of pregnancy [47,48].

Systemic vascular resistance determines the LV afterload. Sex steroids, by direct vasodilation, may reduce vascular resistance. Some authors observed reduced vascular resistance in the luteal phase of the menstrual cycle [29,30]. Gordon et al. assumed that the peripheral vascular tone decrease is a consequence of reduced sensitivity of alpha-adrenergic vascular receptors in this cycle phase [30]. However, we did not notice any significant changes in SVRI during the menstrual cycle. Meta-analysis of studies on peripheral vascular function showed some dilation in macrovascular bed in the late follicular and mid-luteal phase but no change throughout the menstrual cycle in microvascular diameter, which is the primary determinant of vascular resistance [49].

Blood pressure depends on several factors, including the amount of circulating blood, vascular resistance, and the mechanical properties of arteries. During physiological pregnancy, despite the increased plasma volume, blood pressure declines due to increased vascular compliance and decreased vascular resistance [15,28]. Data on blood pressure in healthy women during the menstrual cycle are inconsistent. Chapman et al. showed a decrease in MBP in the luteal phase [29], Adkisson et al. reported the lowest blood pressure in the late follicular phase [18], while others observed BP increase in the luteal phase [50,51]. However, several other studies [16,17], including our results, did not find any differences in blood pressure during the menstrual cycle of healthy women.

The rise in BP values in the luteal phase was explained by the action of progesterone [50] and increased sympathetic activity [50,51]. Progesterone’s mechanism of action is complex. It additionally increases natriuresis and extravascular albumin escape [13,33] and has vasorelaxant properties [8]. The lack of significant changes in blood pressure values across the menstrual cycle may result from the mutual suppression of various actions of progesterone. Moreover, as we discussed earlier, parameters used to describe autonomic control of the cardiovascular system have many limitations, and they are unable to describe this complex issue fully.

### Limitations of the Study

We used cardiac impedance for the hemodynamic assessment, which is less accurate than invasive thermodilution, considered the gold standard clinical method. However, cardiac impedance is non-invasive and is commonly used in many physiological and clinical studies, including those on healthy women [47]. Another limitation is the indirect measurement of total water content in the body. The deuterium dilution analysis, the reference method for this purpose, is expensive as it requires heavy hydrogen, i.e., a non-radioactive and stable hydrogen isotope [52]. Many studies have shown that the estimation of total body water by body impedance is strongly correlated with similar measures by the deuterium dilution method [53,54,55]. In addition, total body impedance is cheap, reproducible, and commonly used in many studies. We focused our study only on healthy women with regular cycles and no hormonal therapy. For this reason, our results should not be extrapolated to women with chronic diseases, e.g., diabetes, hypertension, or obesity, or who are using hormonal contraception. Finally, our study was observational, not aiming at explaining potential mechanisms. Therefore, some considerations are theoretical and might be considered as speculations only.

## 5. Conclusions

It appears that hormonal changes during the menstrual cycle, regardless of the mechanisms, do not produce significant enough changes in cardiovascular function and water management in healthy women of reproductive age.

Haemodynamic indices related to the preload (CI, which equals venous return, TFC, partially PEP, and HR), myocardial contractility (SI, CI, PEP, STR), and afterload (SVRI, DBP) do not change across the menstrual cycle phases. LVET, reflecting the mechanical part of the left ventricular systole, differs statistically by about 5 ms between the mid-luteal and late follicular phases. However, from a clinical standpoint, such a difference is negligible and irrelevant. In addition, other parameters describing myocardial contractility did not differ between these (or any other) phases.

If a woman of reproductive age presents with clinical signs such as increased resting heart rate, blood pressure, or tissue oedema, e.g., swollen legs, these findings are abnormal and require further examination. Such clinical abnormalities must not be considered as expected physiological changes related to various menstrual cycle phases. In other words, ovulation, premenstrual, or other menstrual cycle phases should never explain increased blood pressure or palpitations. On the other hand, if signs of cardiovascular dysfunction or weight/water gain are observed in a young woman using contraceptives, this should raise the suspicion of the side effects of such treatment.

However, the effects of sex hormones on the cardiovascular system and water management are known, for instance, during pregnancy. Nevertheless, the range of sex hormone changes across the menstrual cycle phases in healthy women is narrower than in pregnant women or those on hormonal contraception or other sex hormones prescribed for clinical reasons. Therefore, prospective studies on hormonal drugs in young and potentially healthy women should include body composition analysis and cardiovascular system evaluation methods.

All in all, we demonstrate that in healthy women of reproductive age, hemodynamic parameters and indices of water content do not change during the same menstrual cycle. Whether a woman is at her early follicular, late follicular, or mid-luteal phase, her hemodynamics and body water content are comparable.

## Figures and Tables

**Figure 1 jcm-11-05861-f001:**
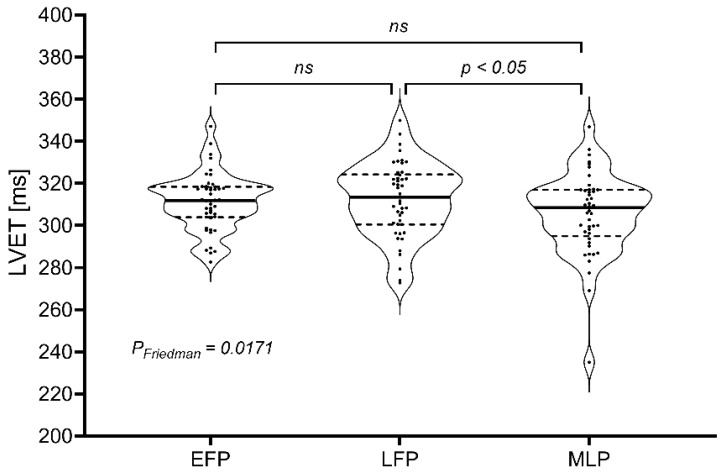
Violin plot for pairwise comparison of LVET measurements in the early follicular phase, late follicular phase and mid-luteal phase of the menstrual cycle. Abbreviations: EFP—early follicular phase; MLP—mid-luteal phase; LFP—late follicular phase; LVET—left ventricle ejection time; ns—not significant.

**Table 1 jcm-11-05861-t001:** The concentration of sex hormones and clinical characteristics of healthy women with regular menstrual cycles.

	Median	25–75 per
Clinical characteristics		
Age [years]	28	26–34
Height [cm]	169.0	164.0–173.0
BMI [kg/m^2^]	21.3	19.8–23.79
WHR	0.7	0.7–0.8
Biochemical profile		
Total cholesterol [mg/dL]	178.1	162.3–202.6
HDL [mg/dL]	68.1	63.6–78.0
LDL [mg/dL]	97.7	83.7–121.6
Triglicerydes [mg/dL]	61.4	48.3–72.0
Glucose [mg/dL]	90.0	84.2–96.2
Insulin [μIU/mL]	5.4	4.4–7.1
TSH [μIU/mL]	1.8	1.2–2.5
Prolactin [ng/mL]	12.4	8.3–18.9
Testosterone [ng/mL]	0.3	0.2–0.4
SHBG [nmol/L]	65.3	48.7–80.3
DHEA-S [μmol/L]	6.1	4.7–7.4
Sex hormones concentration		
E2 in EFP [pg/mL]	40.2	33.6–50.8
FSH in EFP [mIU/mL]	5.8	5.2–6.7
LH in EFP [mIU/mL]	5.2	4.4–6.6
E2 in LFP [pg/mL]	166.7	111.0–290.2
FSH in LFP [mIU/mL]	4.7	3.9–6.1
LH in LFP [mIU/mL]	8.7	7.6–13.1
E2 in MLP [pg/mL]	156.3	118.0–236.8
FSH in MLP [mIU/mL]	5.2	3.2–6.8
LH in MLP [mIU/mL]	3.3	2.6–4.3
Progesterone in MLP [ng/mL]	13.2	10.1–18.2

Abbreviations: BMI—body mass index; DHEA-S—dehydroepiandrosterone sulfate; EFP—early follicular phase; E2—17β estradiol; FSH—follicle-stimulating hormone; LFP—late follicular phase; LH—luteinising hormone; MLP—mid-luteal phase; SHBG—sex hormone-binding globulin; TSH—thyroid stimulating hormone; WHR—waist–hip ratio.

**Table 2 jcm-11-05861-t002:** Haemodynamic profile and indices of body water amount in healthy women during consecutive phases of the same physiological menstrual cycle.

		EFP		LFP		MLP		
		Median	25–75 per	Median	25–75 per	Median	25–75 per	P_Friedman_
**Haemodynamics**	SBP [mmHg]	105.1	101.45–110.79	104.39	101.13–111.45	105.07	97.47–109.04	0.3404
	DBP [mmHg]	64.34	60.54–70.19	65.11	60.99–70.41	62.84	59.82–69.03	0.822
	HR [ud/min]	74.81	65.86–80.89	69.89	65.63–78.38	72.93	65.54–79.64	0.1264
	CI [l/min/m^2^]	3.62	3.25–4.00	3.54	3.13–3.96	3.51	3.09–3.98	0.2144
	SI [ml/m^2^]	50.47	47.58–54.67	49.91	46.07–56.56	49.11	46.42–52.82	0.1789
	SVRI [dyn/s/cm^5^/m^2^]	1604.8	1374.4–1764.3	1558.9	1419.1–1893.5	1682.7	1398.9–1885.9	0.5999
	LVET [ms]	311.9	304.02–318.43	313.52	300.52–323.99	308.4	295.64–316.96	0.0171
	PEP [ms]	97.73	90.16–107.32	98.57	90.86–111.89	97.88	91.73–104.79	0.5999
	STR [%]	0.31	0.29–0.35	0.31	0.29–0.36	0.32	0.30–0.35	0.091
**Water indices**	Body mass [kg]	59	55–65	60	55–65	59.5	55–65	0.7127
	Total body water mass [kg]	32.60	30.80–35.00	32.53	30.65–34.85	32.30	30.30–34.65	0.4134
	Percentage of water [%]	54.60	53.00–57.05	54.50	51.90–57.18	54.17	51.09–56.07	0.5255
	TFC [1/kΩ]	29.27	25.65–32.43	29.77	24.92–32.15	28.44	24.71–33.27	0.7902

Abbreviations: BMI—body mass index; CI—cardiac index; DBP—diastolic blood pressure; EFP—early follicular phase; HR—heart rate; LFP—late follicular phase; LVET—left ventricle ejection time; MLP—mid-luteal phase; PEP—pre-ejection period; SBP—systolic blood pressure; SI—stroke index; STR—systolic period ratio; SVRI—systemic vascular resistance index; TFC—thoracic fluid content; *p* value < 0.05 in bold.

## Data Availability

Not applicable.

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
