# Peer review of "Cardiovascular Function in Different Phases of the Menstrual Cycle in Healthy Women of Reproductive Age"

_jcm, 2022, doi:10.3390/jcm11195861_

Round 1

Reviewer 1 Report

The article entitled Cardiovascular function in different phases of the menstrual 2 cycle in healthy women of reproductive age, is related to the menstrual cycle and how cardiovascular function is modified, however the authors do not find differences in the variables studied.

 The introduction needs to be re-edited because it is necessary to specify the reasons for studying the variables that were used in the research, it is important to emphasize the importance of the research and especially which variables are considered different with respect to the bibliographic evidence.

I consider the methodology adequate, as well as the analysis of the data.

It is necessary to review the discussion because it does not delve into the variables that did not show changes, however, the evidence shows the opposite, for which it is suggested to describe the reason for the differences (resting heart rate, SVRI, blood pressure, etc.)

The conclusion presents limitations due to the results; I suggest increasing the discussion and developing a conclusion based on the reasons for the differences.

Reviewing the bibliography presents a difference with respect to other published articles (italics and bold).

Author Response

The article entitled Cardiovascular function in different phases of the menstrual 2 cycle in healthy women of reproductive age, is related to the menstrual cycle and how cardiovascular function is modified, however the authors do not find differences in the variables studied.

 The introduction needs to be re-edited because it is necessary to specify the reasons for studying the variables that were used in the research, it is important to emphasize the importance of the research and especially which variables are considered different with respect to the bibliographic evidence.

Reply:

Thank you for this critical comment. We have made several changes in the revised manuscript, including the Introduction.

I consider the methodology adequate, as well as the analysis of the data.

Reply:

Thank you for this comment.

It is necessary to review the discussion because it does not delve into the variables that did not show changes, however, the evidence shows the opposite, for which it is suggested to describe the reason for the differences (resting heart rate, SVRI, blood pressure, etc.)

Reply:

Thank you for pointing out the important gaps. In the revised manuscript, we have, hopefully, improved the discussion of our results.

The conclusion presents limitations due to the results; I suggest increasing the discussion and developing a conclusion based on the reasons for the differences.

Reply:

Thank you for this comment. We have modified our conclusion to better reflect clinical meaning of our study.

Reviewing the bibliography presents a difference with respect to other published articles (italics and bold).

Reply: Thank you for brining it to our attention. Proper amendments have been made.

Reviewer 2 Report

Currently I think it is a low-medium paper and MAJOR REVISIONS are necessary.

Here they are POINTS of WEAKNESSES:

-English language to be checked. Sentences such as at lines 35-37 or 293-296 should be re-written with a different fluency.

-Moreover, tables might be greatly ameliorated and limitations of the study better clarified in the text.

-Finally, I'd like to suggest some following articles in order to enlarge discussion/conclusions and/or add to references

PLoS One. 2022 Jun 27;17(6):e0270501. doi: 10.1371/journal.pone.0270501.

Exp Physiol. 2021 Oct;106(10):2031-2037. doi: 10.1113/EP089702.

With my best regards.

Author Response

Currently I think it is a low-medium paper and MAJOR REVISIONS are necessary.

Here they are POINTS of WEAKNESSES:

-English language to be checked. Sentences such as at lines 35-37 or 293-296 should be rewritten with a different fluency.

Reply: Thank you for pointing out the language weaknesses of our manuscript. In a revised manuscript, we have rewritten several parts.  We are  going to have the paper proof-read after the final version has been agreed on.

-Moreover, tables might be greatly ameliorated and limitations of the study better clarified in the text.

Reply: Thank you for these suggestions. We have modified the tables and limitations to improve their clarity.

-Finally, I'd like to suggest some following articles in order to enlarge discussion/conclusions and/or add to references

PLoS One. 2022 Jun 27;17(6):e0270501. doi: 10.1371/journal.pone.0270501.

Exp Physiol. 2021 Oct;106(10):2031-2037. doi: 10.1113/EP089702.

Reply: Thank you for suggesting additional literature to enrich and improve our discussion. Proper changes have been made based on the proposed article PLoS One. 2022 Jun 27;17(6):e0270501. doi: 10.1371/journal.pone.0270501., and additionally position 7. in its references.

However, regarding the second recommendation, we would like to note that the paper of D'Agata et al. considers a racial disparity in macrovascular and microvascular function, not the influence of the menstrual cycle / the concentration of sex hormones on vascular properties. We note papers by Thijssen et al. and Harris et al. suggesting considering the menstrual cycle phases when studying vascular function. However, these recommendations were made in the relatively remote past, and newer reports on this issue are now available. For this reason, we have used another article summarizing the influence of menstrual cycle phases on macro- and microcirculation. Am J Physiol Heart Circ Physiol. 2020 Dec 1;319(6):H1327-H1337. doi: 10.1152/ajpheart.00341.2020.

Round 2

Reviewer 2 Report

Globally, I have appreciated efforts of authors for giving responses to reviewers, so editorial team may considered it for publication even if with low priority in my opinion.